# Partial Multi-Label Learning with Probabilistic Graphical Disambiguation

**Jun-Yi Hang, Min-Ling Zhang**\*
School of Computer Science and Engineering, Southeast University, Nanjing 210096, China
Key Laboratory of Computer Network and Information Integration (Southeast University),
Ministry of Education, China
{hangjy, zhangml}@seu.edu.cn

## Abstract

In partial multi-label learning (PML), each training example is associated with a set of candidate labels, among which only some labels are valid. As a common strategy to tackle PML problem, *disambiguation* aims to recover the ground-truth labeling information from such inaccurate annotations. However, existing approaches mainly rely on heuristics or ad-hoc rules to disambiguate candidate labels, which may not be universal enough in complicated real-world scenarios. To provide a principled way for disambiguation, we make a first attempt to explore the probabilistic graphical model for PML problem, where a directed graph is tailored to infer latent ground-truth labeling information from the generative process of partial multi-label data. Under the framework of stochastic gradient variational Bayes, a unified variational lower bound is derived for this graphical model, which is further relaxed probabilistically so that the desired prediction model can be induced with simultaneously identified ground-truth labeling information. Comprehensive experiments on multiple synthetic and real-world data sets show that our approach outperforms the state-of-the-art counterparts.

## 1 Introduction

Partial multi-label learning aims to induce a multi-label predictor from inaccurately annotated examples, where a set of candidate labels is assigned to each training example but only some of these candidate ones are valid. The problem of PML naturally arises in many real-world applications [39, 51, 52, 20]. For instance, as shown in Figure 1, the data collected from the crowdsourcing platform is inevitable to contain inaccurate supervision information due to potential unreliable annotators, where invalid labels exist in the set of candidate labels.

Formally, let $\mathcal{X} = \mathbb{R}^d$ denote the input space and $\mathcal{Y} = \{l_1, l_2, \dots, l_t\}$ denote the label space with $t$ class labels. A partial multi-label example is denoted as $(\mathbf{x}, S)$, where $\mathbf{x} \in \mathcal{X}$ is its feature vector and $S \subseteq \mathcal{Y}$ is its set of candidate labels. As a basic assumption of PML, the ground-truth labels reside in the candidate label set, i.e. $Y \subseteq S$, and are concealed from the learning algorithm. Given a partial multi-label data set $\mathcal{D} = \{(\mathbf{x}_i, S_i) | 1 \le i \le m\}$, the goal of PML is to derive a multi-label predictor $h : \mathcal{X} \to 2^{\mathcal{Y}}$ which can accurately predict all the ground-truth labels for an unseen instance.

A straightforward strategy for tackling the problem of PML is to simply treat all the candidate labels as valid ones and then apply off-the-shelf multi-label learning approaches [50, 23] to induce the predictor. This naïve strategy is obviously suboptimal as the learning process will be significantly misled by the noisy labels in the candidate label set.

---

\*Corresponding author

37th Conference on Neural Information Processing Systems (NeurIPS 2023).

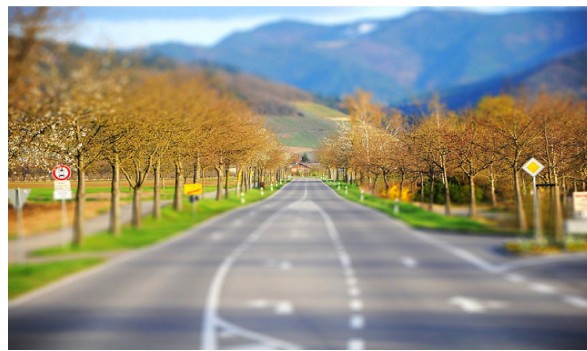

Candidate labels
(valid ones in red)

***tree***
flower
***light***
***mountain***
car
***road***
people

Figure 1: An exemplary partial multi-label learning scenario. Among the set of 7 candidate labels given by crowdsourcing annotators, only 4 labels are valid including *tree*, *light*, *mountain*, and *road*.

Considering that the ground-truth labels are concealed in the candidate label set, it is a common strategy of existing approaches to learn from partial multi-label data by disambiguation, i.e. recovering the ground-truth labeling information from candidate labels. The basic idea of disambiguation is to estimate the labeling confidence for each candidate label from observed inaccurately annotated examples, which reflects how likely the candidate label can be a ground-truth one. For example, some approaches introduce the smoothness assumption to directly elicit the labeling confidence via label propagation [48, 26], label enhancement [31, 41], or cluster assignment [39, 3], etc. While some other works heuristically believe that the noisy labels in the candidate label set are generally sparse, so that the noisy label matrix [30, 29] or the noisy label identifier [35] can be learned with sparsity regularization. Although feasible results have been achieved in these approaches, a potential limitation is that they rely on heuristics or ad-hoc rules for disambiguation, which may not be widely applicable in some challenging scenarios. For example, sparsity on underlying noisy labels can hardly be satisfied in some extremely low-quality annotation scenarios.

To improve this, we explore a principled manner for disambiguation, which is regarded as a task of latent variable inference in this paper. Accordingly, a novel partial multi-label learning approach named PARD, i.e. **P**artial multi-label learning with prob**A**bilistic-g**R**aphical **D**isambiguation, is presented. By regarding the concealed ground-truth labels as latent variable, a directed graphical model is tailored to describe the generative process of partial multi-label data. With the directed graphical model, a unified variational lower bound of the data log-likelihood is derived via variational inference, which allows to disambiguate the candidate labels and induce the prediction model simultaneously. Specifically, the ground-truth labeling information is identified by fitting the generative model of PML data, where the intrinsic structure information of data can be captured for principled disambiguation. While the desired prediction model is optimized with a confidence-smoothed cross entropy loss. Comprehensive experimental studies show that our approach performs better than well-established PML algorithms.

The rest of this paper is organized as follows. Section 2 briefly reviews related works. Section 3 presents details of the proposed PARD approach. Section 4 reports experimental results over a wide range of synthetic and real-world data sets. Section 5 concludes this paper.

## 2 Related Work

Partial multi-label learning is an emerging weakly-supervised learning problem, which has received wide attention in recent years [39, 40, 36]. Here, we would like to first make a brief review on two closely related learning problems, i.e. *multi-label learning* [50, 23] and *partial label learning* [6, 24].

Multi-label learning (MLL) deals with the problem where an example can be associated with multiple labels simultaneously. According to the order of label correlations considered, existing MLL approaches can be grouped into three categories, including *first-order approaches* which treat each class label independently [2, 14], *second-order approaches* which explore label correlations between pairwise class labels [8, 53], and *high-order approaches* which consider label correlations among a subset or the whole set of class labels [32, 44]. With the same goal, both MLL and PML aim to learn a multi-label predictor from multi-semantic objects. Nevertheless, the key difference between these

two learning problems lies in whether the ground-truth labels are accessible or not during the learning process, which also makes PML more challenging than MLL since the supervision information is inaccurate.

Partial label learning (PLL) deals with the problem where each example is associated with multiple candidate labels, among which only one is valid. The problem of PLL can be tackled in several different strategies, such as transforming into well-studied supervised learning problems [49, 34], disambiguating the candidate labels [9, 45, 43], or learning with theoretically consistent algorithms [24, 10], etc. Conceptually speaking, PLL and PML possess similar form of supervision, where false positive labels reside in the candidate label set. However, the task of PML is more complicated than PLL since the desired predictor is a multi-label one in PML instead of a single-label one in PLL and the number of ground-truth labels is further unknown.

To solve the PML problem, the most commonly employed strategy is disambiguation, i.e. recovering the ground-truth labeling information from candidate labels. Existing approaches mainly rely on some heuristics or ad-hoc rules to disambiguate candidate labels, which can be roughly grouped into three categories in terms of the heuristics or rules exploited, including *smoothness assumption-based approaches*, *low-rank constraint-based approaches*, and *sparsity regularization-based approaches*. The first category of approaches rely on smoothness assumption that close instances may share similar labels for disambiguation. As a seminal work, an iterative procedure is proposed in [39] to optimize the labeling confidence of each candidate label and the prediction model alternatively, where the labeling confidence scores are obtained by assuming cluster structures exist in the feature space. Follow-up works employ smoothness assumption to design different disambiguation mechanisms, such as propagating labels in a graph based on instance similarity [48, 26], recovering numerical labels with label enhancement [31, 41, 42], or performing implicit disambiguation by jointly embedding [12]. Low-rank constraint-based approaches turn to the low-rank property for disambiguation. For example, low-rank matrix decomposition is exploited to recover the ground-truth labeling information in [47] and a low-rank subspace is learned in [25] to elicit credible labels from improved subspace representations. While sparsity regularization-based approaches [30, 29, 35] impose sparsity on noises hiding in candidate label set, which facilitate the disambiguation in a roundabout manner. A potential limitation of these approaches is that the above heuristics or ad-hoc rules can be hardly held in some challenging scenarios, which may lead to suboptimal performance of induced prediction model.

Therefore, it is quite natural to consider whether it is possible to perform disambiguation in a principled manner, which is still an underexplored direction with only a few attempts. PML-GAN [46] recovers ground-truth labeling information with an instance reconstruction process, corresponding to a minimax adversarial game to implicitly model instance distribution. While MILI-PML [11] disambiguates candidate labels by maximizing the mutual information between features and identified ground-truth labels. With a totally different methodology, our approach regards disambiguation as a task of latent variable inference, which is formalized in a maximum likelihood framework with a concise variational lower bound of the log-likelihood on observed PML data. As another pioneer work towards this direction, PML-MD [37] devises a meta-learning framework, where disambiguation is performed with the guidance from an accurately annotated validation set. Moving one step forward, we make a first attempt to tackle the problem via modelling the generative process of partial multi-label data with a tailored directed graphical model. Compared with PML-MD, our approach is more flexible since there is no need to include a clean validation set for disambiguation, which may not be easily accessible in many real-world scenarios. We will detail our approach in the next section.

## 3 The PARD Approach

### 3.1 Overview

To tackle the problem of partial multi-label learning, PARD disambiguates the candidate labels and induces the prediction model simultaneously by optimizing towards a unified variational lower bound of the data log-likelihood. In the following content, we will present the tailored directed graphical model which describes the generative process of partial multi-label data. Then, we derive a variational lower bound of the data log-likelihood and further explain how to relate it with disambiguation and predictor induction. Finally, essential implementation issues w.r.t. the optimization procedure are detailed.

Some necessary notations have been introduced in section 1. For notation briefness, a $t$-dimensional indicator vector $\mathbf{s} \in \{0, 1\}^t$ is utilized to denote the set of candidate labels $S$, where $s_k = 1$ indicates $l_k \in S$ and $s_k = 0$ otherwise. Similarly, a $t$-dimensional indicator vector $\mathbf{y} \in \{0, 1\}^t$ is utilized to denote the set of ground-truth labels $Y$.

## 3.2 Probabilistic Graphical Disambiguation Framework

Considering that only the instance and its associated candidate labels can be observed in the partial multi-label training set, we tailor a directed graphical model to describe the generative process of partial multi-label data, which involves an unobserved latent variable $\mathbf{y}$ as the ground-truth labels. The generative process consists of three steps: (1) sample an instance $\mathbf{x}$ from the marginal distribution $p_\theta(\mathbf{x})$; (2) sample a latent $\mathbf{y}$ as the ground-truth labels of the instance $\mathbf{x}$ from the ground-truth class posterior distribution $p_\theta(\mathbf{y}|\mathbf{x})$; (3) corrupt the instance's ground-truth labels to obtain its candidate labels $\mathbf{s}$ by $p_\theta(\mathbf{s}|\mathbf{x}, \mathbf{y})$. Accordingly, the joint probability $p_\theta(\mathbf{x}, \mathbf{y}, \mathbf{s})$ can be factorized as

$$p_\theta(\mathbf{x}, \mathbf{y}, \mathbf{s}) = p_\theta(\mathbf{x})p_\theta(\mathbf{y}|\mathbf{x})p_\theta(\mathbf{s}|\mathbf{x}, \mathbf{y}). \tag{1}$$

Given the partial multi-label training set $\mathcal{D}$, the above directed graphical model can be learned via maximizing the log-likelihood on the observed data. However, it is generally challenging to directly estimate the parameter of a directed graphical model by likelihood maximization due to computational intractability [17]. To make the optimization tractable, the variational lower bound of the log-likelihood[2] is derived as follows

$$\log p_\theta(\mathbf{s}|\mathbf{x}) \geq \mathcal{L}(\mathbf{x}, \mathbf{s}; \theta, \phi) = \mathbb{E}_{q_\phi(\mathbf{y}|\mathbf{x}, \mathbf{s})}[\log \frac{p_\theta(\mathbf{s}|\mathbf{x}, \mathbf{y})p_\theta(\mathbf{y}|\mathbf{x})}{q_\phi(\mathbf{y}|\mathbf{x}, \mathbf{s})}], \tag{2}$$

where $q_\phi(\mathbf{y}|\mathbf{x}, \mathbf{s})$ is the introduced variational posterior to approximate the true posterior $p_\theta(\mathbf{y}|\mathbf{x}, \mathbf{s})$. $\mathcal{L}(\mathbf{x}, \mathbf{s}; \theta, \phi)$ is the derived variational lower bound as a surrogate loss function of the log-likelihood, which can be rewritten as

$$\mathcal{L}(\mathbf{x}, \mathbf{s}; \theta, \phi) = \mathbb{E}_{q_\phi(\mathbf{y}|\mathbf{x}, \mathbf{s})}[\log p_\theta(\mathbf{s}|\mathbf{x}, \mathbf{y})] - KL[q_\phi(\mathbf{y}|\mathbf{x}, \mathbf{s})||p_\theta(\mathbf{y}|\mathbf{x})], \tag{3}$$

where $KL[\cdot||\cdot]$ denotes the KL-divergence between two distributions. Here, the meaning of the ground-truth class posterior distribution $p_\theta(\mathbf{y}|\mathbf{x})$ is comprehensible, which is actually the desired prediction model[3]. While the variational posterior $q_\phi(\mathbf{y}|\mathbf{x}, \mathbf{s})$ (a.k.a. inference model) attempts to disambiguate the candidate label set by inferring the most probable ground-truth labels from which the candidate labels could have been corrupted given the instance $\mathbf{x}$. Correspondingly, $p_\theta(\mathbf{s}|\mathbf{x}, \mathbf{y})$ (a.k.a. generative model) corrupts the identified ground-truth labels to recover the observed candidate labels.

To move one step further for comprehending the learning behavior of the above three models, we reformulate the variational lower bound via unfolding the KL-divergence term

$$\begin{aligned}\mathcal{L}(\mathbf{x}, \mathbf{s}; \theta, \phi) = &\mathbb{E}_{q_\phi(\mathbf{y}|\mathbf{x}, \mathbf{s})}[\log p_\theta(\mathbf{s}|\mathbf{x}, \mathbf{y})] + H[q_\phi(\mathbf{y}|\mathbf{x}, \mathbf{s})] \\ &+ \mathbb{E}_{q_\phi(\mathbf{y}|\mathbf{x}, \mathbf{s})}[\log p_\theta(\mathbf{y}|\mathbf{x})],\end{aligned} \tag{4}$$

where $H[\cdot]$ denotes the entropy of a distribution. In the above objective, the first two terms present an entropy-regularized autoencode process (w.r.t. $\mathbf{s}$), with which the intrinsic structure information of data can be captured and utilized for disambiguation. While the last term is actually a cross entropy loss for inducing the prediction model with identified ground-truth labeling information $q_\phi(\mathbf{y}|\mathbf{x}, \mathbf{s})$. By optimizing all these terms in a unified variational lower bound, the models can gradually capture the underlying generative process of partial multi-label data, so that the candidate label set is disambiguated and the desired prediction model is induced simultaneously.

## 3.3 Additional Training Techniques

In this section, we present additional techniques that allow the derived variational lower bound to be optimized efficiently.

---

[2]We omit the optimization for the marginal distribution on $\mathbf{x}$, as it is not the focus of our work. Detailed derivation process can be found in Appendix B.

[3]With a little abuse of notations, a distribution, e.g. $p_\theta(\mathbf{y}|\mathbf{x})$, is also called a model which actually denotes the model to parameterize the distribution, when the context is clear. All the distributions involved in Eq. (3) are instantiated as multivariate Bernoulli distributions and are parameterized by neural networks.

**Algorithm 1** Pseudocode of the Optimization Procedure for PARD
***
**Inputs:** Training set $\mathcal{D} = \{(\mathbf{x}_i, S_i) | 1 \le i \le m\}$, batch size $b$ and maximal iteration $T$.
**Process:**
  1: Initialize model parameters $\theta, \phi$;
  2: **for** $t = 1 : T$ **do**
  3:     Sample a batch of training examples $\mathcal{B} = \{(\mathbf{x}_{i_k}, S_{i_k}) | 1 \le k \le b\}$ from training set $\mathcal{D}$;
  4:     Compute unbiased estimator of the variational lower bound on $\mathcal{B}$ by Eq. (6) and Eq. (7);
  5:     Update model parameters $\theta, \phi$ via gradient ascent.
  6: **end for**
**Outputs**: Model parameters $\theta, \phi$.
***

**Continuous Relaxing with Gumbel-Softmax Trick.**    It is quite challenging to compute the first expectation term in Eq. (3). Actually, it has a high complexity of $\mathcal{O}(2^t)$ to analytically compute the first expectation term in Eq. (3), since the ground-truth labels $\mathbf{y}$ can take $2^t$ different values. In addition, estimating it by Monte Carlo sampling is non-differentiable, since sampling from discrete variational posterior $q_\phi(\mathbf{y}|\mathbf{x}, \mathbf{s})$ prevents gradients from flowing through the inference model.

To circumvent this problem, we exploit *Gumbel-Softmax trick* [13, 27] to smooth the sampling process so that the expectation term can be efficiently estimated by Monte Carlo sampling in a differentiable manner. Specifically, the Bernoulli variable $\mathbf{y}$ is relaxed by the binary Concrete variable $\mathbf{c} \sim BinConcrete(\mathbf{p}, \tau)$, which is a continuous alternative with the reparameterization form as

$$\mathbf{c} = \frac{1}{1 + \exp[-(\log \boldsymbol{\alpha} + \mathbf{l})/\tau]}, \tag{5}$$

where $\boldsymbol{\alpha} = \frac{\mathbf{p}}{1-\mathbf{p}}$ and $\mathbf{p}$ is the parameter of the multivariate Bernoulli distribution $q_\phi(\mathbf{y}|\mathbf{x}, \mathbf{s})$. $\mathbf{l}$ is a sampling from Logistic distribution and $\tau > 0$ is a temperature parameter. In the limit $\tau \to 0$, a binary Concrete variable smoothly converges to its Bernoulli counterpart. With the above sampling trick, an unbiased estimation of the expectation term can be obtained and gradients w.r.t. the distribution parameter $\mathbf{p}$ (or w.r.t. the parameter of the inference model equivalently) are well-defined by the chain rule

$$\mathbb{E}_{q_\phi(\mathbf{y}|\mathbf{x}, \mathbf{s})}[\log p_\theta(\mathbf{s}|\mathbf{x}, \mathbf{y})] \approx \frac{1}{L} \sum_{i=1}^{L} \log p_\theta(\mathbf{s}|\mathbf{x}, \mathbf{c}^{(i)})$$
$$where \quad \mathbf{c}^{(i)} \sim BinConcrete(\mathbf{p}, \tau). \tag{6}$$

**Closed-Form Solution of the KL-Divergence Term.**    The KL-divergence term in Eq. (3) can not be integrated analytically in general cases. To make it tractable, we exploit mean-field approximation technique and derive a closed-form solution of the KL-divergence term as follows

$$KL[q_\phi(\mathbf{y}|\mathbf{x}, \mathbf{s}) || p_\theta(\mathbf{y}|\mathbf{x})] = \sum_{k=1}^{t} p_\phi^{y_k} \log \frac{p_\phi^{y_k}}{p_\theta^{y_k}} + (1 - p_\phi^{y_k}) \log \frac{1 - p_\phi^{y_k}}{1 - p_\theta^{y_k}}, \tag{7}$$

where $p_\phi^{y_k} = q_\phi(y_k = 1|\mathbf{x}, \mathbf{s})$ and $p_\theta^{y_k} = p_\theta(y_k = 1|\mathbf{x})$. Detailed derivation process can be found in Appendix C.

With these training techniques, the derived variational lower bound can be straightforwardly optimized with standard stochastic gradient methods. Algorithm 1 summarizes the pseudocode of the optimization procedure, where the candidate labels and the prediction model are updated simultaneously.

## 4   Experiments

### 4.1   Experimental Setup

#### 4.1.1   Data Sets

For comprehensive performance evaluation, five real-world and a number of synthetic PML data sets are employed in this paper. Table 1 summarizes detailed characteristics of each data set. Specifically,

Table 1: Characteristics of the experimental data sets. The fisrt five ones are real-world PML data sets and the last five ones are multi-label data sets employed to generate synthetic PML data sets.

| Dataset | #Examples | #Features | #Class Labels | Cardinality | Domain |
|---------|-----------|-----------|---------------|-------------|--------|
| YeastBP | 6139 | 6139 | 217 | 5.54 | Biology[1] |
| YeastCC | 6139 | 6139 | 50 | 1.35 | Biology[1] |
| YeastMF | 6139 | 6139 | 39 | 1.01 | Biology[1] |
| Music_emotion | 6833 | 98 | 11 | 2.42 | Music[1] |
| Music_style | 6839 | 98 | 10 | 1.44 | Music[1] |
| corel5k | 5000 | 499 | 374 | 3.52 | Images[2] |
| rcv1-s1 | 6000 | 944 | 101 | 2.88 | Text[2] |
| Corel16k-s1 | 13766 | 500 | 153 | 2.86 | Images[2] |
| iaprtc12 | 19627 | 1000 | 291 | 5.72 | Images[3] |
| espgame | 20770 | 1000 | 268 | 4.69 | Images[3] |

[1] `http://palm.seu.edu.cn/zhangml/`
[2] `http://mulan.sourceforge.net/datasets.html`
[3] `http://lear.inrialpes.fr/people/guillaumin/data.php`

the first five data sets are real-world PML data sets, where candidate labels are collected from web users and manually examined to specify the ground-truth labels. While the last five data sets, including *corel5k*, *rcv1-s1*, *Corel16k-s1*, *iaprtc12* and *espgame*, are multi-label data sets.

Following [35], a synthetic PML data set is generated from one multi-label data set by adding random labeling noise. Accordingly, for data sets with more than 100 class labels, we filter out their rare labels to keep the 15 most frequent labels and remove instances without relevant labels. Then, we randomly flip the irrelevant labels of an instance until the number of irrelevant labels in the set of candidate labels equals to $\gamma\%$ number of ground-truth labels. We vary the value of $\gamma$ in the range of $\{100, 150, 200, 250\}$, so that a total of $5 \times 4 = 20$ synthetic PML data sets are generated for experimental studies.

#### 4.1.2 Evaluation Metrics

Five widely-used evaluation metrics for multi-label learning are employed to evaluate the performance of each approach, including *Average precision*, *Hamming loss*, *One-error*, *Coverage* and *Ranking loss*. Detailed definitions on these metrics can be found in [50].

#### 4.1.3 Implementation Details

The inference model $q_\phi(\mathbf{y}|\mathbf{x}, \mathbf{s})$ and the generative model $p_\theta(\mathbf{s}|\mathbf{x}, \mathbf{y})$ are instantiated by fully-connected neural networks with ReLU activations, where the hidden dimensionalities are set to $[256; 512; 256]$ and $[256; 512]$ respectively. For fair comparison with existing PML approaches, the prediction model is implemented as a linear model. To compute the objective function in Eq. (3), a trade-off parameter $\alpha$ is introduced for the KL-divergence term and Monte Carlo sampling with sampling number $L = 1$ is conducted to estimate the first expectation term, where the temperature parameter $\tau = 2/3$ as suggested by [27]. In the following experiments, we set $\alpha \geq 1$ so that the objective function is still a valid lower bound of the data log-likelihood. For network optimization, Adam with a batch size of 128, weight decay of $10^{-4}$, momentums of 0.999 and 0.9 is employed. In this paper, all experiments are conducted on one V100 GPU.

### 4.2 Comparative Studies

PARD[4] is compared against six well-established PML approaches with parameter configurations suggested in respective literatures:

- FPML [47]: FPML employs the low-rank approximation of the instance-label association matrix to estimate the labeling confidence and then trains multi-label predictor. [$\lambda_1 = 0.1, \lambda_2 = 1, \lambda_3 = 10$]

---

[4] Code package of PARD is publicly available at `http://palm.seu.edu.cn/zhangml/files/PARD.rar`.

Table 2: Predictive performance of each comparing approach (mean±std. deviation) in terms of *Average precision*, where •/○ indicates whether PARD is significantly superior/inferior to one comparing approach via paired $t$-test at 0.05 significance level. ↑ (↓) indicates the larger (smaller) the value, the better the performance. Best results are shown in boldface.

| Data sets | γ% | *Average precision* ↑ | | | | | | |
|---|---|---|---|---|---|---|---|---|
| | | FPML | PARVLS | PML-NI | PML-MD | UPML-HL | UPML-RL | PARD |
| YeastBP | | 0.202±0.013• | 0.059±0.003• | **0.430±0.018**○ | 0.310±0.016• | 0.392±0.020• | 0.170±0.009• | 0.417±0.015 |
| YeastCC | | 0.375±0.019• | 0.145±0.009• | 0.610±0.022• | 0.527±0.027• | 0.596±0.022• | 0.459±0.021• | **0.622±0.020** |
| YeastMF | | 0.279±0.020• | 0.114±0.007• | 0.471±0.022• | 0.422±0.027• | 0.451±0.016• | 0.329±0.023• | **0.491±0.028** |
| Music_emotion | | 0.571±0.013• | 0.603±0.011• | 0.600±0.012• | 0.646±0.011 | **0.656±0.012**○ | 0.639±0.012• | 0.650±0.007 |
| Music_style | | 0.702±0.014• | 0.720±0.012• | 0.733±0.012• | 0.716±0.012• | 0.734±0.010• | 0.703±0.012• | **0.742±0.009** |
| corel5k | 100 | 0.485±0.014• | 0.391±0.017• | 0.473±0.016• | 0.491±0.019• | 0.516±0.015• | 0.514±0.015• | **0.533±0.013** |
| | 150 | 0.478±0.013• | 0.408±0.011• | 0.453±0.013• | 0.479±0.015• | 0.514±0.015• | 0.512±0.015• | **0.523±0.013** |
| | 200 | 0.474±0.013• | 0.395±0.021• | 0.438±0.012• | 0.472±0.011• | 0.503±0.009• | 0.511±0.013 | **0.516±0.011** |
| | 250 | 0.470±0.013• | 0.391±0.022• | 0.426±0.011• | 0.470±0.010• | 0.502±0.013• | 0.507±0.014 | **0.513±0.012** |
| rcv1-s1 | 100 | 0.682±0.010• | 0.520±0.021• | 0.693±0.014• | 0.678±0.010• | 0.712±0.013• | 0.689±0.007• | **0.720±0.008** |
| | 150 | 0.678±0.011• | 0.513±0.022• | 0.664±0.013• | 0.656±0.011• | 0.705±0.013• | 0.690±0.008• | **0.716±0.010** |
| | 200 | 0.673±0.010• | 0.495±0.021• | 0.649±0.010• | 0.648±0.010• | 0.704±0.012• | 0.685±0.005• | **0.710±0.010** |
| | 250 | 0.666±0.011• | 0.488±0.016• | 0.630±0.011• | 0.640±0.011• | 0.693±0.015• | 0.672±0.009• | **0.706±0.009** |
| Corel16k-s1 | 100 | 0.535±0.010• | 0.430±0.010• | 0.518±0.011• | 0.522±0.011• | 0.539±0.012• | 0.525±0.007• | **0.551±0.012** |
| | 150 | 0.529±0.008• | 0.423±0.012• | 0.499±0.011• | 0.512±0.011• | 0.532±0.011• | 0.522±0.009• | **0.545±0.012** |
| | 200 | 0.523±0.010• | 0.404±0.012• | 0.487±0.013• | 0.507±0.012• | 0.504±0.011• | 0.519±0.009• | **0.536±0.014** |
| | 250 | 0.514±0.011• | 0.389±0.011• | 0.476±0.013• | 0.501±0.011• | 0.507±0.014• | 0.516±0.008• | **0.531±0.014** |
| iaprtc12 | 100 | 0.603±0.006• | 0.591±0.008• | 0.599±0.009• | 0.600±0.008• | 0.601±0.009• | 0.563±0.008• | **0.621±0.011** |
| | 150 | 0.600±0.006• | 0.585±0.008• | 0.580±0.010• | 0.583±0.010• | 0.597±0.007• | 0.559±0.007• | **0.615±0.009** |
| | 200 | 0.597±0.008• | 0.574±0.008• | 0.563±0.008• | 0.569±0.009• | 0.600±0.008• | 0.554±0.008• | **0.610±0.008** |
| | 250 | 0.588±0.009• | 0.561±0.008• | 0.543±0.009• | 0.560±0.009• | 0.582±0.008• | 0.547±0.006• | **0.601±0.009** |
| espgame | 100 | 0.498±0.007• | 0.472±0.007• | 0.478±0.008• | 0.491±0.008• | 0.501±0.008• | 0.466±0.009• | **0.515±0.008** |
| | 150 | 0.496±0.007• | 0.456±0.007• | 0.459±0.007• | 0.481±0.007• | 0.509±0.008 | 0.471±0.006• | **0.511±0.009** |
| | 200 | 0.495±0.007• | 0.445±0.007• | 0.446±0.005• | 0.473±0.007• | 0.475±0.012• | 0.466±0.006• | **0.508±0.008** |
| | 250 | 0.489±0.009• | 0.427±0.009• | 0.431±0.007• | 0.467±0.007• | 0.490±0.008• | 0.460±0.006• | **0.501±0.009** |

- PARVLS [48]: A two-step approach which elicits credible labels via label propagation and then induces multi-label predictor by virtual label splitting. [$k = 10$, $\alpha = 0.95$, and $thr = 0.9$]

- PML-NI [35]: PML-NI performs disambiguation by learning a noisy label identifier with sparsity regularization. [$\beta = 0.5$, $\gamma = 0.5$, and search for $\lambda \in \{1, 10, 100\}$]

- PML-MD [37]: PML-MD disambiguates between ground-truth and noisy labels in a meta-learning fashion and learns multi-label predictor with a confidence-weighted ranking loss.

- UPML-HL [36]: A disambiguation-free approach which directly induces the prediction model by optimizing an unbiased estimator of the Hamming loss.

- UPML-RL [36]: A disambiguation-free approach which directly induces the prediction model by optimizing an unbiased estimator of the Ranking loss.

Following [37], we take out 10% examples in each data set as hold-out validation set, so that PML-MD can use this clean validation set to perform meta-disambiguation. The remaining 90% examples are randomly splitted into training set and test set with a ratio of 9:1 for training and evaluation repectively. During training, PML-MD performs meta-learning on the noisy training set and the clean validation set. While other approaches including ours only employ this validation set to search hyperparameters. In PARD, the trade-off paramter $\alpha$ is searched in $\{1, 2, 5, 10, 20, 50, 100\}$. For fair comparison, all deep approaches adopt a linear model to instantiate the predictor and share the same optimizer with the learning rate searched in {1e-3,3e-3,1e-2,3e-2}. For each data set, we repeat the random splitting process for ten times and record the average predicitve performance across ten training/test trials.

Due to page limit, Table 2 and Table 3 report detailed experimental results in terms of *Average precision* and *Ranking loss*. Results on other metrics can be found in Appendix A. Furthermore, the *Wilcoxon signed-ranks test* [33] at 0.05 significance level is conducted to analyze whether PARD achieves statistically superior performance to other comparing approaches. Table 4 summarizes the $p$-value statistics on each evaluation metric. Based on these results, it is impressive to observe that:

Table 3: Predictive performance of each comparing approach (mean±std. deviation) in terms of *Ranking loss*, where •/○ indicates whether PARD is significantly superior/inferior to one comparing approach via paired $t$-test at 0.05 significance level. ↑ (↓) indicates the larger (smaller) the value, the better the performance. Best results are shown in boldface.

| Data sets | $\gamma\%$ | Ranking loss ↓ | | | | | | |
|---|---|---|---|---|---|---|---|---|
| | | FPML | PARVLS | PML-NI | PML-MD | UPML-HL | UPML-RL | PARD |
| YeastBP | | 0.338±0.007• | 0.482±0.005• | 0.197±0.012• | 0.218±0.008• | 0.222±0.015• | 0.309±0.011• | **0.175±0.010** |
| YeastCC | | 0.305±0.016• | 0.480±0.010• | 0.156±0.017• | 0.190±0.014• | 0.167±0.017• | 0.212±0.015• | **0.135±0.017** |
| YeastMF | | 0.373±0.019• | 0.533±0.010• | 0.226±0.018• | 0.235±0.014• | 0.239±0.023• | 0.286±0.017• | **0.192±0.015** |
| Music_emotion | | 0.274±0.012• | 0.252±0.009• | 0.249±0.010• | 0.231±0.010 | **0.223±0.009**○ | 0.236±0.011• | 0.228±0.007 |
| Music_style | | 0.160±0.010• | 0.150±0.007• | 0.139±0.008 | 0.143±0.007• | 0.138±0.007 | 0.149±0.007• | **0.137±0.009** |
| corel5k | 100 | 0.255±0.013• | 0.357±0.020• | 0.280±0.012• | 0.251±0.011• | 0.236±0.014• | 0.229±0.013• | **0.223±0.012** |
| | 150 | 0.260±0.013• | 0.327±0.011• | 0.296±0.015• | 0.260±0.014• | 0.235±0.014 | 0.235±0.013 | **0.233±0.010** |
| | 200 | 0.263±0.011• | 0.338±0.021• | 0.312±0.013• | 0.264±0.012• | 0.254±0.010• | **0.240±0.012** | 0.242±0.010 |
| | 250 | 0.268±0.012• | 0.347±0.024• | 0.322±0.013• | 0.267±0.011• | 0.257±0.013• | **0.242±0.013** | 0.245±0.010 |
| rcv1-s1 | 100 | 0.101±0.004• | 0.222±0.013• | 0.106±0.007• | 0.093±0.005• | 0.086±0.005• | 0.087±0.003• | **0.076±0.003** |
| | 150 | 0.102±0.005• | 0.240±0.020• | 0.126±0.006• | 0.105±0.005• | 0.089±0.004• | 0.089±0.004• | **0.079±0.003** |
| | 200 | 0.107±0.006• | 0.262±0.022• | 0.138±0.005• | 0.111±0.004• | 0.092±0.007• | 0.093±0.003• | **0.084±0.004** |
| | 250 | 0.114±0.006• | 0.277±0.017• | 0.153±0.005• | 0.115±0.005• | 0.099±0.006• | 0.100±0.004• | **0.090±0.003** |
| Corel16k-s1 | 100 | 0.219±0.006• | 0.289±0.007• | 0.247±0.009• | 0.221±0.008• | 0.212±0.006• | 0.216±0.006• | **0.205±0.006** |
| | 150 | 0.226±0.006• | 0.300±0.011• | 0.262±0.009• | 0.228±0.006• | 0.224±0.007• | 0.220±0.007• | **0.212±0.007** |
| | 200 | 0.235±0.007• | 0.329±0.011• | 0.274±0.011• | 0.232±0.006• | 0.250±0.009• | 0.223±0.006 | **0.221±0.009** |
| | 250 | 0.245±0.009• | 0.369±0.010• | 0.286±0.012• | 0.238±0.007• | 0.249±0.011• | 0.228±0.006• | **0.223±0.008** |
| iaprtc12 | 100 | 0.189±0.004• | 0.207±0.005• | 0.206±0.006• | 0.192±0.005• | 0.199±0.007• | 0.210±0.004• | **0.180±0.006** |
| | 150 | 0.191±0.004• | 0.212±0.005• | 0.222±0.007• | 0.203±0.005• | 0.203±0.006• | 0.211±0.005• | **0.186±0.005** |
| | 200 | 0.196±0.004• | 0.227±0.006• | 0.239±0.006• | 0.214±0.005• | 0.203±0.006• | 0.216±0.005• | **0.191±0.005** |
| | 250 | 0.204±0.006• | 0.241±0.005• | 0.257±0.006• | 0.222±0.006• | 0.218±0.007• | 0.221±0.006• | **0.199±0.005** |
| espgame | 100 | 0.252±0.005• | 0.281±0.006• | 0.285±0.004• | 0.258±0.006• | 0.256±0.006• | 0.277±0.008• | **0.241±0.005** |
| | 150 | 0.256±0.006• | 0.300±0.005• | 0.302±0.005• | 0.266±0.006• | 0.248±0.006• | 0.265±0.006• | **0.245±0.006** |
| | 200 | 0.258±0.006• | 0.313±0.006• | 0.314±0.005• | 0.272±0.006• | 0.277±0.008• | 0.269±0.005• | **0.248±0.006** |
| | 250 | 0.266±0.006• | 0.330±0.007• | 0.329±0.007• | 0.274±0.006• | 0.269±0.007• | 0.275±0.006• | **0.255±0.005** |

Table 4: Summary of the Wilcoxon signed-ranks test for PARD against other comparing approaches at 0.05 significance level. $p$-values are shown in the brackets.

| PARD against | FPML | PARVLS | PML-NI | PML-MD | UPML-HL | UPML-RL |
|---|---|---|---|---|---|---|
| *Average precision* | **win** [1.2e-5] | **win** [1.2e-5] | **win** [1.8e-5] | **win** [1.2e-5] | **win** [1.7e-5] | **win** [1.2e-5] |
| *Hamming loss* | **win** [8.2e-3] | **win** [4.6e-5] | **win** [2.1e-5] | **win** [8.5e-5] | tie [7.0e-1] | **win** [3.5e-5] |
| *One-error* | **win** [1.2e-5] | **win** [1.2e-5] | **win** [4.6e-5] | **win** [1.2e-5] | **win** [1.8e-4] | **win** [1.2e-5] |
| *Coverage* | **win** [1.2e-5] | **win** [1.2e-5] | **win** [1.4e-5] | **win** [1.2e-5] | **win** [2.4e-5] | **win** [4.0e-5] |
| *Ranking loss* | **win** [1.2e-5] | **win** [1.2e-5] | **win** [1.2e-5] | **win** [1.2e-5] | **win** [2.0e-5] | **win** [2.5e-5] |

- Across all evaluation metrics, PARD achieves the best performance in 82.4% cases over all the 25 data sets.

- As shown in Table 4, PARD achieves statistically better performance against other approaches which rely on heuristics or ad-hoc rules for disambiguation. The superior performance of PARD is consistent across almost all real-world data sets and synthetic data sets under varied noise levels, which provides a strong evidence for the effectiveness of probabilistic graphical disambiguation to facilitate partial multi-label learning.

- Meantime, PARD significantly outperforms PML-MD which also disambiguates PML data in a principled manner. Note that PML-MD requires an accurately annotated validation set to perform disambiguation during the learning process. The superior performance of PARD against PML-MD indicates that PARD is an effective and more flexible approach for principled disambiguation.

## 4.3 Further Analyses

### 4.3.1 Ablation Study

To validate that the disambiguation process in PARD is effective for facilitating partial multi-label learning, we implement a baseline which induces prediction model with cross-entropy loss by directly treating all the candidate labels as valid ones. As shown in Figure 2, a clear performance degradation

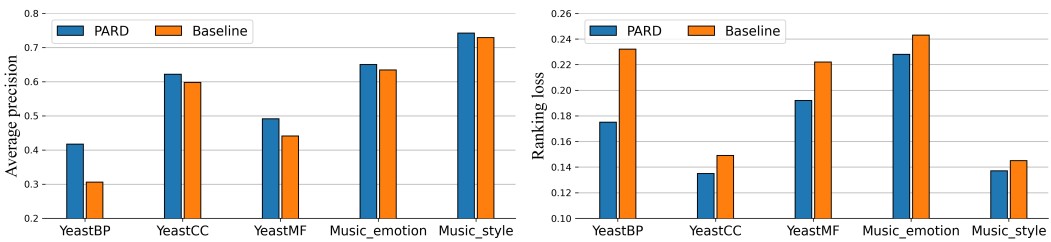

Figure 2: Predictive performance of PARD and the baseline in terms of *Average precision* and *Ranking loss* on real-world data sets.

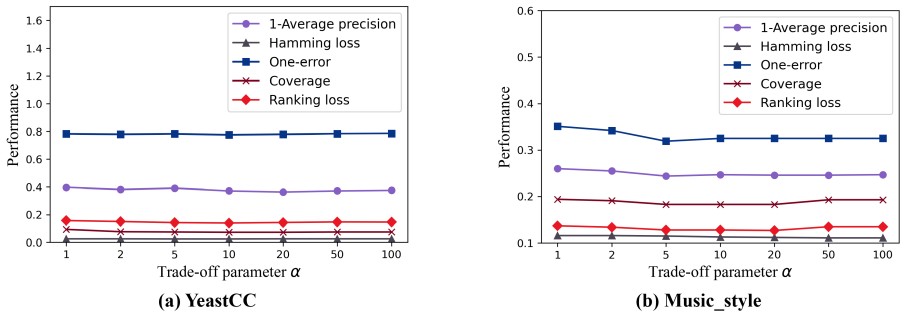

**(a) YeastCC**                    **(b) Music_style**

Figure 3: Validation performance of PARD with varying trade-off parameter $\alpha$.

is witnessed in the baseline model, which demonstrates the disambiguation process induced from the variational lower bound can facilitate partial multi-label learning.

### 4.3.2 Parameter Sensitivity

Figure 3 gives some illustrative examples on how the performance of PARD changes when the value of the trade-off parameter $\alpha$ changes. The performance of PARD is relatively stable as the value of $\alpha$ changes within a reasonable range, which is a desirable property when deploying PARD in real-world applications. Similar results can be observed on other data sets.

### 4.3.3 Running Time Comparision

For PARD, the training phase only involves a pair of forward and backward computations among the inference, generative and prediction models, which is efficient thanks to the training techniques presented in section 3.3. Figure 4 illustrates the empirical training and test time of each comparing approach, which shows that PARD is competitive with exisiting approaches in time overhead.

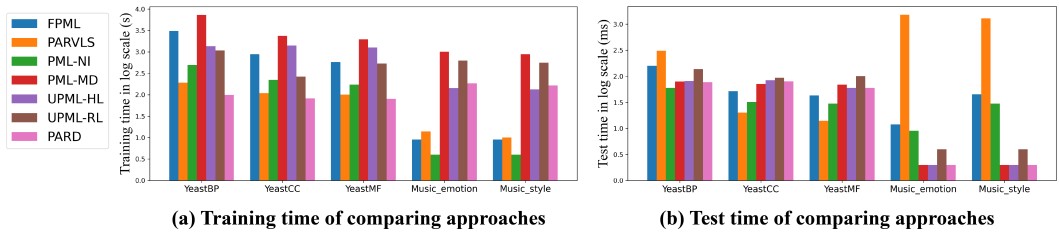

**(a) Training time of comparing approaches**          **(b) Test time of comparing approaches**

Figure 4: Running time (training/test) of each comparing approach on real-world data sets. For histogram illustration, the $y$-axis corresponds to the logarithm of running time.

## 5 Limitation

Our primary focus is on partial multi-label learning in this paper. However, there are other weakly-supervised multi-label learning problems that receive increasing attention from related community. The probabilistic graphical disambiguation framework of PARD currently has not been tested on these problems, such as learning with partial labels which deals with incompletely annotated data [7, 19, 1, 4], learning with single positive label which aims to induce an accurate multi-label predictor from multi-label data annotated with only one positive labels [5, 28, 15, 38, 16, 22], and learning with general noisy labels where noise may exist in every label of inaccurately annotated data [18, 21, 36]. In addition, we do not implement PARD and investigate its behavior in large model environment, which are also important to practitioners. We will explore these points in the future.

## 6 Conclusion

In this paper, an attempt towards principled disambiguation for partial multi-label learning is presented, where a directed graphical model is tailored to describe the generative process of partial multi-label data. By maximizing the data log-likelihood on given partial multi-label data set with a unified surrogate objective derived by variational inference, PARD achieves to disambiguate the candidate label set and induce the prediction model simultaneously. Comprehensive experiments show the superiority of our approach. Since disambiguation lies in the heart of partial multi-label learning, we hope that PARD will encourage more future researches to explore alternative implementations for principled disambiguation.

## Acknowledgments

The authors wish to thank the anonymous reviewers for their helpful comments and suggestions. This work was supported by the National Science Foundation of China (62225602), and the Big Data Computing Center of Southeast University.

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
