# Partial Multi-Label Learning with Probabilistic Graphical Disambiguation
# Supplementary Material

**Jun-Yi Hang, Min-Ling Zhang**[*]

School of Computer Science and Engineering, Southeast University, Nanjing 210096, China
Key Laboratory of Computer Network and Information Integration (Southeast University),
Ministry of Education, China
{hangjy, zhangml}@seu.edu.cn

## Appendix A
## More Experimental Results for Comparative Studies

Table A.1, A.2 and A.3 report detailed experimental results in terms of *Coverage*, *One-error* and *Hamming loss*, which are not covered in the *Comparative Studies* part of the main body due to page limit. It can be observed that our PARD achieves consistently superior performance to well-established PML approaches.

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

# Appendix B
## Derivation of The Variational Lower Bound

The variational lower bound of the log-likelihood (i.e. Eq. (2) in the main body) is derived as follows

$$\log p_\theta(\mathbf{s}|\mathbf{x}) = \log \int p_\theta(\mathbf{s}, \mathbf{y}|\mathbf{x})d\mathbf{y}$$

$$= \log \int p_\theta(\mathbf{s}|\mathbf{x}, \mathbf{y})p_\theta(\mathbf{y}|\mathbf{x})d\mathbf{y}$$

$$= \log \int q_\phi(\mathbf{y}|\mathbf{x}, \mathbf{s}) \frac{p_\theta(\mathbf{s}|\mathbf{x}, \mathbf{y})p_\theta(\mathbf{y}|\mathbf{x})}{q_\phi(\mathbf{y}|\mathbf{x}, \mathbf{s})} d\mathbf{y}$$

$$\geq \mathbb{E}_{q_\phi(\mathbf{y}|\mathbf{x},\mathbf{s})}[\log \frac{p_\theta(\mathbf{s}|\mathbf{x}, \mathbf{y})p_\theta(\mathbf{y}|\mathbf{x})}{q_\phi(\mathbf{y}|\mathbf{x}, \mathbf{s})}]$$

$$= \mathcal{L}(\mathbf{x}, \mathbf{s}; \theta, \phi).$$

# Appendix C
## Derivation of The KL-Divergence Term's Closed-Form Solution

With mean-field approximation technique, closed-form solution of the KL-divergence term can be derived as follows

$$KL[q_\phi(\mathbf{y}|\mathbf{x}, \mathbf{s})||p_\theta(\mathbf{y}|\mathbf{x})]$$

$$= \mathbb{E}_{q_\phi(\mathbf{y}|\mathbf{x},\mathbf{s})}[\log q_\phi(\mathbf{y}|\mathbf{x}, \mathbf{s}) - \log p_\theta(\mathbf{y}|\mathbf{x})]$$

$$= \mathbb{E}_{q_\phi(\mathbf{y}|\mathbf{x},\mathbf{s})}[\sum_{k=1}^{t} \log q_\phi(y_k|\mathbf{x}, \mathbf{s}) - \sum_{k=1}^{t} \log p_\theta(y_k|\mathbf{x})]$$

$$= \sum_{k=1}^{t} \mathbb{E}_{q_\phi(\mathbf{y}|\mathbf{x},\mathbf{s})}[\log \frac{q_\phi(y_k|\mathbf{x}, \mathbf{s})}{p_\theta(y_k|\mathbf{x})}]$$

$$= \sum_{k=1}^{t} \mathbb{E}_{q_\phi(y_k|\mathbf{x},\mathbf{s})}[\log \frac{q_\phi(y_k|\mathbf{x}, \mathbf{s})}{p_\theta(y_k|\mathbf{x})}]$$

$$= \sum_{k=1}^{t} KL[q_\phi(y_k|\mathbf{x}, \mathbf{s})||p_\theta(y_k|\mathbf{x})]$$

$$= \sum_{k=1}^{t} p_\phi^{y_k} \log \frac{p_\phi^{y_k}}{p_\theta^{y_k}} + (1 - p_\phi^{y_k}) \log \frac{1 - p_\phi^{y_k}}{1 - p_\theta^{y_k}}.$$

Althouth mean-field approximation would restrict model capacity, it is a routine in VAE-related literatures [17] to make graphical model tractable. A natural direction for future work is to investigate whether it is possible to compute the KL-divergence term without mean-field approximation.