# OpenReview forum: "Partial Multi-Label Learning with Probabilistic Graphical Disambiguation"
_NeurIPS.cc/2023/Conference — NeurIPS 2023 poster_

### Official Review · Reviewer_GJDR · 2023-06-29

**Soundness:** 3 good
**Presentation:** 3 good
**Contribution:** 2 fair
**Rating:** 5
**Confidence:** 3

**Summary:**

The paper focuses on a specific problem called partial multi-label learning (PML), which is an emerging weakly-supervised learning problem. PML involves inaccurate annotations, where each training example is associated with a set of candidate labels, among which only some labels are valid. The paper proposes a probabilistic graphical model for PML and develops a principled approach for disambiguating candidate labels. The approach is evaluated through comprehensive experiments on multiple synthetic and real-world datasets, showing superior performance compared to existing methods.

**Strengths:**

1. The paper presents a novel approach called PARD for partial multi-label learning, which aims to recover the groundtruth labeling information from inaccurate annotations.
2. PARD uses a directed graphical model to describe the generative process of partial multi-label data and a unified variational lower bound derived by variational inference to maximize the data log-likelihood on given partial multi-label data set.
3. PARD achieves to disambiguate the candidate label set and induce the prediction model simultaneously, and comprehensive experiments show the superiority of this approach.

**Weaknesses:**

1. PARD provides a principled way for disambiguation, which is the process of recovering the groundtruth labeling information from inaccurate annotations. This can be useful in many real-world scenarios where the annotations are not perfect.However, the number of labels required may not meet the requirements of the actual task, and further work can be done in this regard.
2. The use of a directed graphical model and variational inference in PARD can be extended to other machine learning tasks beyond partial multi-label learning, where probabilistic modeling and inference are required. But the construction of modeling and inference rules is very dependent on the nature of the specific task, is there a good transfom method to take advantage of the previous work?

**Questions:**

1. Any idea about how to explore alternative implementations for principled disambiguation?
2. The proposed approach PARD outperforms the state-of-the-art counterparts on multiple synthetic and real-world datasets, demonstrating its effectiveness in improving the accuracy of partial multi-label learning tasks. But I'd like to see how this approach behaves in a large model environment, or how it might be combined
3. The results of the comprehensive experiments on multiple synthetic and real-world datasets show that PARD outperforms the state-of-the-art counterparts, which indicates that this approach can be used to improve the performance of many practical applications that involve partial multi-label learning. But why does this model achieve the best experimental results on all tasks? What about other tasks?

---

> ### Author Rebuttal · Authors · 2023-08-08
>
> ------
> Thank you for your reviews and insightful comments. The following are our responses to the **Questions** and **Weaknesses**:
>
> ------
> 1. *Any idea about how to explore alternative implementations for principled disambiguation?*
>
> * **Response 1**: Thanks to the comments. We believe that new ideas on alternative implementations for principled disambiguation can be inspired from related problems of partial multi-label learning (PML). For example, some methods in noisy label learning, e.g. those based on sample selection or transition matrix, could be adapted for tackling PML problem.
>
>   ------
> 2. *The proposed approach PARD outperforms the state-of-the-art counterparts on multiple synthetic and real-world datasets, demonstrating its effectiveness in improving the accuracy of partial multi-label learning tasks. But I'd like to see how this approach behaves in a large model environment, or how it might be combined.*
>
> * **Response 2**: Thanks to the comments. Actually, PARD can be implemented flexibly with different network structures. To combine with large models, a feasible way is to use the large model as a shared feature extractor and attach three heads (i.e. inference model, generative model and prediction model) on top of the large model. Then, the whole model can be trained with the unified variational lower bound in Eq. (3). However, training a good large model is not an easy work due to limited time and may involve many tricks. We believe it is interesting to investigate the behavior of PARD in the large model environment and will explore it in the future.
>
>   ------
> 3. *The results of the comprehensive experiments on multiple synthetic and real-world datasets show that PARD outperforms the state-of-the-art counterparts, which indicates that this approach can be used to improve the performance of many practical applications that involve partial multi-label learning. But why does this model achieve the best experimental results on all tasks? What about other tasks?*
>
> * **Response 3**: Thanks to the comments. Existing approaches mainly rely on heuristics or ad-hoc rules to disambiguate candidate labels. These heuristics or rules can be too strong to be satisfied in real-world scenarios. For example, some approaches rely on sparsity property for disambiguation, although the underlying noisy labels in the set of candidate labels may not be sparse in some extremely low-quality annotation scenarios. While our approach has no such requirement, which directly disambiguates and induces prediction model by maximizing the log-likelihood on observed data. Therefore, it can handle diversified PML tasks more flexibly and outperform existing approaches in multiple benchmark data sets.
>
>   ------
> 4. *The use of a directed graphical model and variational inference in PARD can be extended to other machine learning tasks beyond partial multi-label learning, where probabilistic modeling and inference are required. But the construction of modeling and inference rules is very dependent on the nature of the specific task, is there a good transform method to take advantage of the previous work?*
>
> * **Response 4**: Thanks to the comments. We agree with the reviewer that the directed graphical model and variational inference are useful tools to tackle various machine learning tasks. And the graphical model and inference rules do need some flexible customization according to the properties of the specific task. For example, in PARD, we tailor the graphical model to describe the generative process of partial multi-label data and exploit additional techniques (e.g. Gumbel-Softmax relaxing) to make inference efficient. In our opinion, it is exactly the point making these tools fascinating to machine learning community, since it allows to introduce inductive bias or expert knowledge from humans.
>
> ------

---

### Official Review · Reviewer_65gX · 2023-07-03

**Soundness:** 2 fair
**Presentation:** 3 good
**Contribution:** 2 fair
**Rating:** 2
**Confidence:** 5

**Summary:**

In the manuscript, the authors adopt the probabilistic graphical model to solve the disambiguation problem in partial multi-label learning. Specifically, a directed graph is tailored to describe the generative process of partial multi-label data, and a partial multi-label learning approach named PARD is proposed. PARD attempts to disambiguate the candidate label set and induce the prediction model simultaneously by maximizing the data log-likelihood with a unified surrogate objective derived by variational inference. The authors conduct extensive experiments to validate the effectiveness of the proposed method.

**Strengths:**

The idea to use probabilistic graphical model for PML problem is novel. The manuscript is easy to follow.

**Weaknesses:**

The motivation is not clear. Why do authors apply probabilistic graphical model for disambiguation in PML?

The authors claim that existing approaches are not universal enough in complicated real-world scenarios; however, in real world applications, the data distribution is hard to follow the assumptions claimed when directly tailoring the graph model, which leads to the unreasonability of the proposed approach.

The technical part is not solid, e.g., it is not clear to write the variational posterior q in the subscript of expectation through all the equations.


-------
Thanks the author for the reply. However, I still have the concerns about this paper.

1, The motivation is quite weak.  Why do we need directed graphical model for PML? What are the limitations of existing methods? Why the proposed directed graphical model achieve better performance than other methods? The author did not show the necessity of introducing the directed graphical model for PML.  Moreover, The authors make frequent assertions in the abstract and introduction about existing approaches being heuristic and ad-hoc as the SOTA limitation that motivates the paper. However, the fact is that existing work exactly solves the heuristic or ad-hoc problem.

2,The key issue of this paper is the lack of the theoretical and technical contribution. The paper directly applies the very known technique graphical model for PML. I can not find any new theoretical and technical contributions in this paper.

Therefore, I vote for the rejection.

**Questions:**

The motivation is not clear. Why do authors apply probabilistic graphical model for disambiguation in PML?

The authors claim that existing approaches are not universal enough in complicated real-world scenarios; however, in real world applications, the data distribution is hard to follow the assumptions claimed when directly tailoring the graph model, which leads to the unreasonability of the proposed approach.

The technical part is not solid, e.g., it is not clear to write the variational posterior q in the subscript of expectation through all the equations.

**Limitations:**

N\A

---

> ### Author Rebuttal · Authors · 2023-08-08
>
> ------
> Thank you for your reviews and insightful comments. The following are our responses to the **Questions**:
>
> ------
> 1. *The motivation is not clear. Why do authors apply probabilistic graphical model for disambiguation in PML?*
>
> * **Response 1**: Thanks to the comments. The probabilistic graphical model is known to be an effective tool to infer latent variables based on observed variables. For example, in representation disentanglement literature [1-3], probabilistic graphical model has been widely investigated to recover latent factors of variation in the data which inherently control what the observed data looks like following the data generative process. Inspired by it, we regard the disambiguation as a task of latent variable inference in our work, thus leading to a first attempt to introduce probabilistic graphical model for PML. Under a maximum likelihood framework, we aim to infer underlying ground-truth labeling information by fitting the generative model of PML data. Empirical results on multiple synthetic and real-world data sets show that the probabilistic graphical model is a promising tool for tackling PML problem. We will make it clearer in the revised version.
>
>   [1] Higgins, I., et al. Beta-VAE: Learning basic visual concepts with a constrained variational framework. In Proceedings of the 5th International Conference on Learning Representations.
>
>   [2] Kim, H. and Mnih, A. Disentangling by factorising. In Proceedings of the 35th International Conference on Machine Learning.
>
>   [3] Van Steenkiste, S., et al. Are disentangled representations helpful for abstract visual reasoning? In Advances in Neural Information Processing Systems 32.
>
>   ------
> 2. *The authors claim that existing approaches are not universal enough in complicated real-world scenarios; however, in real world applications, the data distribution is hard to follow the assumptions claimed when directly tailoring the graph model, which leads to the unreasonability of the proposed approach.*
>
> * **Response 2**: Thanks to the comments. Existing approaches mainly rely on some heuristics or ad-hoc rules to disambiguate candidate labels. These heuristics or rules can be too strong to be satisfied in real-world scenarios. For example, some approaches rely on sparsity property for disambiguation, although the underlying noisy labels in the set of candidate labels may not be sparse in some extremely low-quality annotation scenarios. Instead, our approach only makes mild assumption on the generative process of PML data (the first paragraph in section 3.2, pp.3) and keeps enough capacity in the hypothesis space of the graphical model to approximate the real data distribution.
>
>   ------
> 3. *The technical part is not solid, e.g., it is not clear to write the variational posterior q in the subscript of expectation through all the equations.*
>
> * **Response 3**: Thanks to the comments. Actually, we follow the writing style in [4] to write the variational posterior in the subscript of expectation. For example, $\mathbb{E}\_{q\_\phi(\mathbf{y}|\mathbf{x},\mathbf{s})} [\log p\_\theta(\mathbf{s}|\mathbf{x},\mathbf{y})]$ denotes $\int q\_\phi(\mathbf{y}|\mathbf{x},\mathbf{s}) \cdot \log p\_\theta(\mathbf{s}|\mathbf{x},\mathbf{y}) d\mathbf{y}$​. We will make it clearer in the revised version.
>
>   [4] Kingma, D. P. and Welling, M. Auto-encoding variational bayes. In Proceedings of the 2nd International Conference on Learning Representations.
>
> ------

---

> > ### Comment · Reviewer_65gX · 2023-08-22
> > **major issue for rejection**
> >
> > Thanks for the authors' response. The author did not address my concerns. The motivation is still not convincing. The authors make frequent assertions in the abstract and introduction about existing approaches being heuristic and ad-hoc as the SOTA limitation that motivates the paper, which is also point out by Reviewer mCzx. However, the fact is that existing work in the cited reference [7] "Understanding partial multi-label learning via mutual information" exactly solves the heuristic or ad-hoc problem.
> >
> > After my initial review, I find that this manuscript is largely overlapped with the reference [7]; the motivation, and the idea of objective function derivation share many resemblances in the theoretical part. But the manuscript only cited it, without any discussion or comparison w.r.t. the difference.
> >
> > As also pointed out by Reviewer mCzx that the technical idea of disambiguation shares resemblances with the PML-GAN paper.
> >
> > To conclude, there is no enough novelty and contribution either from both theoretical or technical part for the NeurIPS community. I insist rejection.

---

> > > ### Author Response · Authors · 2023-08-22
> > > **Thanks**
> > >
> > > Thank you again for the valuable comments. We would like to emphasize that our work makes the first attempt to employ a directed graphical model for tackling PML problem. And the methodology to identify the ground-truth labeling information is totally different between MILI-PML and PARD. In MILI-PML, ground-truth labels are identified from candidate labels by maximizing the mutual information between features and subset of identified ground-truth labels. While in PARD, the identification process of ground-truth labels is regarded as a task of latent variable inference, with a directed graphical model tailored to disambiguate candidate labels and induce prediction model simultaneously. These differences in the methodology also reflect in the derivation process of objective function and employed derivation techniques. We will include the discussion between MILI-PML and PARD in the revised version.

---

> > > > ### Comment · Reviewer_65gX · 2023-08-22
> > > > **major issue still exist**
> > > >
> > > > Thanks author for the reply. However, my concern still do not well addressed. The reason is listed as follows.
> > > >
> > > > 1, The author says "We would like to emphasize that our work makes the first attempt to employ a directed graphical model for tackling PML problem''.  Why do we need directed graphical model for PML? What are the limitations of existing methods? Why the proposed directed graphical model achieve better performance than other methods? The author did not show the necessity of introducing the directed graphical model for PML. Therefore, I think the motivation is very weak and the author did not address my concerns.
> > > >
> > > > 2, The theoretical and technical contribution is far below the NeurIPS average bar. The paper directly applies the very known technique graphical model for PML. I can not find any new  theoretical and technical contributions. Moreover,  the technical idea of disambiguation shares resemblances with the PML-GAN paper. The author did not address these concerns.
> > > >
> > > > 3, One of my last comments is that " The authors make frequent assertions in the abstract and introduction about existing approaches being heuristic and ad-hoc as the SOTA limitation that motivates the paper, which is also point out by Reviewer mCzx. However, the fact is that existing work in the cited reference [7] "Understanding partial multi-label learning via mutual information" exactly solves the heuristic or ad-hoc problem." The author did not address these concerns.
> > > >
> > > > 4,  Reviewer mCzx comments that I find that the idea of disambiguation shares resemblances with the PML-GAN paper. The author replies to Reviewer mCzx that "we do believe that the methodologies and implementations between PML-GAN and our approach are different. And our approach has solider theoretical support from probabilistic graphical modelling." However, the author did not show any difference about the methodologies and implementations between PML-GAN and their method. Moreover, the author says that "our approach has solider theoretical support from probabilistic graphical modelling". I can not find any theoretical support in this paper.
> > > >
> > > > Overall, I still think this paper is far below the NeurIPS average bar. Therefore, I recommend the rejection.

---

### Official Review · Reviewer_fZq4 · 2023-07-03

**Soundness:** 3 good
**Presentation:** 3 good
**Contribution:** 3 good
**Rating:** 7
**Confidence:** 3

**Summary:**

The paper introduces PARD , a new approach for addressing the problem of partial multilabel learning. PARD uses a probabilistic graphical model to recover ground-truth labeling information. By leveraging the framework of stochastic gradient variational Bayes, PARD simultaneously identifies labeling information and induces a prediction model. Experimental  results on synthetic and real-world datasets demonstrate that PARD outperforms existing methods, providing a more effective and universal solution for partial multi-label learning.

**Strengths:**

1. The article presents a novel approach for addressing the problem of partial multi-label learning. It introduces a probabilistic graphical model and leverages stochastic gradient variational Bayes to simultaneously identify labeling information and induce a prediction model.
2.The article is well-structured and effectively communicates the proposed approach, experimental setup, and results. The authors provide sufficient details, making it easier to understand and replicate the methodology.
3.The article provides extensive experimental results on both synthetic and real-world datasets. It demonstrates that PARD outperforms existing methods, indicating its effectiveness in improving the accuracy of partial multi-label learning tasks.

**Weaknesses:**

Many papers on weakly supervised multi-label have been proposed recently, the author are  encouraged to provide a detailed discussion and if possible, comparing with them.

[1] Y. Kim, J. M. Kim, Z. Akata, and J. Lee, “Large loss matters in weakly supervised multi-label classification,” in Proceedings of the IEEE/CVF Conference on Computer Vision and Pattern Recognition, 2022, pp. 14 156–14 165.
[2] S. Rajeswar, P. Rodriguez, S. Singhal, D. Vazquez, and A. Courville, “Multi-label iterated learning for image classification with label ambiguity,” in Proceedings of the IEEE/CVF Conference on Computer Vision and Pattern Recognition, 2022, pp. 4783–4793.

**Questions:**

1.The derivation from Eq1 to Eq2 can be more detailed explanation.

**Limitations:**

The article deals with the situation of partial multi label.  It seems that for general weakly supervised multi-label
learning like label noise or single positive label, the proposed method is not generalizable.

---

> ### Author Rebuttal · Authors · 2023-08-08
>
> ------
> Thank you for your reviews and insightful comments. The following are our responses to the **Questions** and **Weaknesses**:
>
> ------
> 1. *The derivation from Eq1 to Eq2 can be more detailedly explained.*
>
> * **Response 1**: Thanks to the comments. Detailed derivation process has been provided in Appendix B. We will make it clearer in the revised version.
>
>   ------
> 2. *Many papers on weakly supervised multi-label have been proposed recently, the authors are encouraged to provide a detailed discussion and if possible, comparing with them.*
>
> * **Response 2**: Thanks to the suggestions. We will include the discussion on recent progress in more settings of weakly supervised multi-label learning (e.g. learning with single positive label, learning with partial labels, and learning with general noisy labels) in the revised version. And we will list it as an important future work to investigate whether the probabilistic graphical model can be adapted to tackle these interesting weakly supervised learning problems.
>
> ------

---

> > ### Comment · Reviewer_fZq4 · 2023-08-19
> >
> > I'm satisfied with the authors' response. I would keep my score.

---

> > > ### Author Response · Authors · 2023-08-19
> > > **Thanks**
> > >
> > > Thank you again for the valuable comments. We will check the manuscript again and add the discussion in the revised version.

---

### Official Review · Reviewer_mCzx · 2023-07-04

**Soundness:** 3 good
**Presentation:** 3 good
**Contribution:** 3 good
**Rating:** 6
**Confidence:** 2

**Summary:**

This paper presents a novel, systematic approach to partial multi-label learning (PML) and label disambiguation using a probabilistic graphical model and the stochastic gradient variational Bayes framework. The authors offer a reliable method tested extensively on synthetic and real-world datasets. The results underscore its performance and demonstrate its potential for future PML advancements.

**Strengths:**

The paper presents an interesting and effective technique for PML using graphical models. The approach is explained clearly, although incorporating an illustration of the graphical model would enhance the paper. The extensive experiments validate the method's effectiveness. The overall structure and writing style are lucid and easy to comprehend.

**Weaknesses:**

- The authors make frequent assertions about existing approaches being heuristic and ad-hoc without providing any explanation or justification. Furthermore, there are similar approaches to the one proposed that are not adequately discussed or compared in the paper [1].

- The related work section could be improved by including brief descriptions of some recent related methods.

- The experimental section lacks a comparison with more recent and state-of-the-art papers such as MILI-PML [2] and PML-GAN [1].

[1] Yan, Yan, and Yuhong Guo. "Adversarial partial multi-label learning with label disambiguation." Proceedings of the AAAI Conference on Artificial Intelligence. Vol. 35. No. 12. 2021.

[2] Gong, Xiuwen, Dong Yuan, and Wei Bao. "Understanding partial multi-label learning via mutual information." Advances in Neural Information Processing Systems 34 (2021): 4147-4156.

**Questions:**

- Could the authors provide further clarification regarding the omission of the marginal distribution on x?

- Why is there no discussion and comparison with the Yan and Guo paper [1], which also explores similar dependencies in label disambiguation?

- While the paper claims to outperform state-of-the-art approaches, it does not specify which ones are considered state-of-the-art in the comparisons.


[1] Yan, Yan, and Yuhong Guo. "Adversarial partial multi-label learning with label disambiguation." Proceedings of the AAAI Conference on Artificial Intelligence. Vol. 35. No. 12. 2021.

**Limitations:**

limitations were not explicitly addressed

---

> ### Author Rebuttal · Authors · 2023-08-08
>
> ------
> Thank you for your reviews and insightful comments. The following are our responses to the **Questions** and **Weaknesses**:
>
> ------
> 1. *Could the authors provide further clarification regarding the omission of the marginal distribution on $\mathbf{x}$?*
>
> * **Response 1**: Thanks to the comments. The whole log-likelihood on observed PML data $(\mathbf{x},\mathbf{s})$ can be factorized as follows
>   $$
>   \log p_\theta(\mathbf{x},\mathbf{s})=\log p_{\theta_1}(\mathbf{x})+\log p_{\theta_2}(\mathbf{s}|\mathbf{x}),
>   $$
>   where $\theta,\theta_1,\theta_2$ parametrize distributions $p_\theta(\mathbf{x},\mathbf{s}),p_{\theta_1}(\mathbf{x}),p_{\theta_2}(\mathbf{s}|\mathbf{x})$ respectively, and $\theta=\\{\theta_1,\theta_2\\}$.
>
>   Therefore, maximizing the whole log-likelihood is actually equivalent to maximizing the marginal distribution on $\mathbf{x}$ and the conditional log-likelihood separately
>   $$
>   \max_\theta \log p_\theta(\mathbf{x},\mathbf{s})\equiv \max_{\theta_1,\theta_2} \log p_{\theta_1}(\mathbf{x})+\log p_{\theta_2}(\mathbf{s}|\mathbf{x}).
>   $$
>   Since the focus of our work is to induce the prediction model $p(\mathbf{y}|\mathbf{x})$ which only involves in the conditional likelihood $p_{\theta_2}(\mathbf{s}|\mathbf{x})$ according to Eq. (1), we omit the optimization for the marginal distribution on $\mathbf{x}$.
>
>   ------
> 2. *Why is there no discussion and comparison with the Yan and Guo paper, which also explores similar dependencies in label disambiguation?*
>
> * **Response 2**: Thanks to the comments. In PML-GAN, the disambiguation process is regularized by instance reconstruction, which corresponds to a minimax adversarial game to implicitly model instance distribution $p(\mathbf{x})$. While in our approach, the disambiguation process is learned with a unified variational lower bound of the log-likelihood on observed PML data $(\mathbf{x},\mathbf{s})$, which provides solid theoretical support for its effectiveness. Though both PML-GAN and PARD perform disambiguation in an autoencode process, the underlying methodologies and implementations are different. We will include these discussions in the revised version. However, codes of PML-GAN are not publicly available and we fail to reproduce their results due to limited time. Instead, we further compare our approach with two SOTA algorithms uPMLHL and uPMLRL which are published in 2023 [1]. Representative results (in terms of Average precision$\uparrow$) are shown in the following table and detailed results will be included in the revised version. Our approach still shows superior performance.
>
>   |          Data sets          |       uPMLHL        |     uPMLRL      |        PARD         |
>   | :------------------------- | :-----------------: | :-------------: | :-----------------: |
>   |           YeastBP           |   0.392$\pm$0.020   | 0.170$\pm$0.009 |   **0.417$\pm$0.015**   |
>   |           YeastCC           |   0.596$\pm$0.022   | 0.459$\pm$0.021 | **0.622$\pm$0.020** |
>   |           YeastMF           |   0.451$\pm$0.016   | 0.329$\pm$0.023 | **0.491$\pm$0.028** |
>   |        Music_emotion        | **0.656$\pm$0.012** | 0.639$\pm$0.012 |   0.650$\pm$0.007   |
>   |         Music_style         |   0.734$\pm$0.010   | 0.703$\pm$0.012 | **0.742$\pm$0.009** |
>   |   corel5k ($\gamma$%=250)   |   0.502$\pm$0.013   | 0.507$\pm$0.014 | **0.513$\pm$0.012** |
>   |   rcv1-s1 ($\gamma$%=250)   |   0.693$\pm$0.015   | 0.672$\pm$0.009 | **0.706$\pm$0.009** |
>   | Corel16k-s1 ($\gamma$%=250) |   0.507$\pm$0.014   | 0.516$\pm$0.008 | **0.531$\pm$0.014** |
>   |  iaprtc12 ($\gamma$%=250)   |   0.582$\pm$0.008   | 0.547$\pm$0.006 | **0.601$\pm$0.009** |
>   |   espgame ($\gamma$%=250)   |   0.490$\pm$0.008   | 0.460$\pm$0.006 | **0.501$\pm$0.009** |
>
>   [1] Xie, M.-K. and Huang, S.-J. CCMN: A general framework for learning with class-conditional multi-label noise. TPAMI.
>
>   ------
> 3. *While the paper claims to outperform state-of-the-art approaches, it does not specify which ones are considered state-of-the-art in the comparisons.*
>
> * **Response 3**: Thanks to the comments. In comparative studies, we have compared our approach against 6 existing PML algorithms, four of which are published after 2020. Furthermore, we include two more recent and SOTA algorithms uPMLHL and uPMLRL published in 2023. Comparative results can be found in the table of **Response 2**. Our approach still shows superior performance.
>
>   ------
> 4. *The authors make frequent assertions about existing approaches being heuristic and ad-hoc without providing any explanation or justification.*
>
> * **Response 4**: Thanks to the comments. Existing approaches mainly rely on some heuristics or ad-hoc rules to disambiguate candidate labels. Generally speaking, these approaches can be roughly grouped into three categories in terms of the heuristics or rules exploited, including smoothness assumption-based approaches [2, 3], sparsity regularization-based approaches [4, 5], and low-rank constraint-based approaches [6, 7]. These heuristics or rules can be too strong to be satisfied in real-world scenarios, which may lead to suboptimal performance of induced prediction model. For example, some approaches rely on sparsity property for disambiguation, although the underlying noisy labels in the set of candidate labels may not be sparse in some extremely low-quality annotation scenarios. We will make it clearer in the revised version.
>
>   [2] Wang, H., et al. Discriminative and correlative partial multi-label learning. IJCAI.
>
>   [3] Zhang, M.-L. and Fang, J.-P. Partial multi-label learning via credible label elicitation. TPAMI.
>
>   [4] Xie, M.-K. and Huang, S.-J. Partial multi-label learning with noisy label identification. TPAMI.
>
>   [5] Sun, L., et al. Global-local label correlation for partial multi-label learning. TMM.
>
>   [6] Yu, G., et al. Feature-induced partial multi-label learning. ICDM.
>
>   [7] Sun, L., et al. Partial multi-label learning by low-rank and sparse decomposition. AAAI.
>
> ------

---

> > ### Comment · Reviewer_mCzx · 2023-08-17
> >
> > I appreciate your response to the questions. I'm inclined to maintain my current score, as I find that the idea of disambiguation shares resemblances with the PML-GAN paper. Furthermore, the main text lacks comprehensive discussion of the limitations within the existing research.

---

> > > ### Author Response · Authors · 2023-08-17
> > > **Thanks**
> > >
> > > Thank you again for the valuable comments. PML-GAN is indeed an inspiring work, but we do believe that the methodologies and implementations between PML-GAN and our approach are different. And our approach has solider theoretical support from probabilistic graphical modelling. We will include further discussions on the differences and potential limitations (e.g. unknown generalization ability to more general weakly-supervised settings and behavior in large model environment) of our approach in the revised version.

---

### Official Review · Reviewer_uhwx · 2023-07-07

**Soundness:** 2 fair
**Presentation:** 3 good
**Contribution:** 2 fair
**Rating:** 5
**Confidence:** 5

**Summary:**

This paper attempts to tackle the Partial Multi-Label Learning task by employing the probabilistic graph method. The proposed method employs the directed graph to model the generative process of partial multi-label data, then derives a variational lower bound objective with a gumbel-softmax trick and closed form of KL-divergence. Empirical results on serval benchmarks demonstrate the effectiveness of the proposed method.

**Strengths:**

-	This paper is well-written and quite easy to follow.
-	The idea of tackling partial multi-label learning with probabilistic graphical model seems to be interesting.
-	Empirical results demonstrate the effectiveness of the proposed method.

**Weaknesses:**

-	The motivation of this paper is not clear. In this paper, the authors attempt to employ the probabilistic graphical model to perform a principled way for disambiguation and learn the classifier. However, why can the probabilistic graphical model achieve principled disambiguation? The simple entropy-regularized autoencode process seems to be not enough.
-	Some baseline methods in recent years are missing in the comparison experiment. You can consider some baselines published in 2022 and 2023, such as Global [1] and PAKS [2]. And these two papers are mentioned in the references, but the relevant citations are missing in the text.
-	What is the baseline model in the ablation study? Do you mean that it learns the prediction model with cross-entropy loss by directly treating all the candidate labels as valid ones?
-	In formula 6, there is an extra '] '.

[1] Lijuan Sun, Songhe Feng, Jun Liu, Gengyu Lyu, and Congyan Lang. Global-local label correlation for partial multi-label learning. IEEE Transactions on Multimedia, 24:581–593, 2022.

[2] Gengyu Lyu, Songhe Feng, and Yidong Li. Partial multi-label learning via probabilistic graph matching mechanism. In Proceedings of the 26th ACM SIGKDD International Conference on Knowledge Discovery and Data Mining, pages 105–113, Virtual Event, 2020.

**Questions:**

-	Please see the weakness for details.

**Limitations:**

There seem to be no limitations or potential negative social impact of this paper.

---

> ### Author Rebuttal · Authors · 2023-08-08
>
> ------
> Thank you for your reviews and insightful comments. The following are our responses to the **Questions**:
>
> ------
> 1. *The motivation of this paper is not clear. In this paper, the authors attempt to employ the probabilistic graphical model to perform a principled way for disambiguation and learn the classifier. However, why can the probabilistic graphical model achieve principled disambiguation? The simple entropy-regularized autoencode process seems to be not enough.*
>
> * **Response 1**: Thanks to the comments. The probabilistic graphical model is known to be an effective tool to infer latent variables based on observed variables. For example, in representation disentanglement literature [1-3], probabilistic graphical model has been widely investigated to recover latent factors of variation in the data which inherently control what the observed data looks like following the data generative process. Inspired by it, we regard the disambiguation as a task of latent variable inference in our work, thus leading to a first attempt to introduce probabilistic graphical model for partial multi-label learning (PML). Under a maximum likelihood framework, we aim to infer underlying ground-truth labeling information by fitting the generative model of PML data. Empirical results on multiple synthetic and real-world data sets show that the probabilistic graphical model is a promising tool for tackling PML problem. We will make it clearer in the revised version.
>
>   [1] Higgins, I., et al. Beta-VAE: Learning basic visual concepts with a constrained variational framework. In Proceedings of the 5th International Conference on Learning Representations.
>
>   [2] Kim, H. and Mnih, A. Disentangling by factorising. In Proceedings of the 35th International Conference on Machine Learning.
>
>   [3] Van Steenkiste, S., et al. Are disentangled representations helpful for abstract visual reasoning? In Advances in Neural Information Processing Systems 32.
>
>   ------
> 2. *Some baseline methods in recent years are missing in the comparison experiment. You can consider some baselines published in 2022 and 2023, such as Global and PAKS. And these two papers are mentioned in the references, but the relevant citations are missing in the text.*
>
> * **Response 2**: Thanks to the suggestions. We have included more recent approaches for comparative studies. Global and PAKS are two important works and have already been cited in the Related Work section (pp. 3, citation indices [16] and [13] respectively). We will add further discussions on these two works in the revised version. However, their codes are not publicly available and we fail to reproduce their results due to limited time. Instead, we further compare our approach with two SOTA algorithms uPMLHL and uPMLRL which are published in 2023 [4]. Representative results (in terms of Average precision$\uparrow$) are shown in the following table and detailed results will be included in the revised version. Our approach still shows superior performance.
>
>   |      Data sets      |       PML-NI        |     PML-MD      |       uPMLHL        |     uPMLRL      |        PARD         |
>   | :----------------- | :-----------------: | :-------------: | :-----------------: | :-------------: | :-----------------: |
>   |       YeastBP       | **0.430$\pm$0.018** | 0.310$\pm$0.016 |   0.392$\pm$0.020   | 0.170$\pm$0.009 |   0.417$\pm$0.015   |
>   |       YeastCC       |   0.610$\pm$0.022   | 0.527$\pm$0.027 |   0.596$\pm$0.022   | 0.459$\pm$0.021 | **0.622$\pm$0.020** |
>   |       YeastMF       |   0.471$\pm$0.022   | 0.422$\pm$0.027 |   0.451$\pm$0.016   | 0.329$\pm$0.023 | **0.491$\pm$0.028** |
>   |    Music_emotion    |   0.600$\pm$0.012   | 0.646$\pm$0.011 | **0.656$\pm$0.012** | 0.639$\pm$0.012 |   0.650$\pm$0.007   |
>   |     Music_style     |   0.733$\pm$0.012   | 0.716$\pm$0.012 |   0.734$\pm$0.010   | 0.703$\pm$0.012 | **0.742$\pm$0.009** |
>   |   corel5k ($\gamma$%=250)   |   0.426$\pm$0.011   | 0.470$\pm$0.019 |   0.502$\pm$0.013   | 0.507$\pm$0.014 | **0.513$\pm$0.012** |
>   |   rcv1-s1 ($\gamma$%=250)   |   0.630$\pm$0.011   | 0.640$\pm$0.011 |   0.693$\pm$0.015   | 0.672$\pm$0.009 | **0.706$\pm$0.009** |
>   | Corel16k-s1 ($\gamma$%=250) |   0.476$\pm$0.013   | 0.501$\pm$0.011 |   0.507$\pm$0.014   | 0.516$\pm$0.008 | **0.531$\pm$0.014** |
>   |  iaprtc12 ($\gamma$%=250)   |   0.543$\pm$0.009   | 0.560$\pm$0.009 |   0.582$\pm$0.008   | 0.547$\pm$0.006 | **0.601$\pm$0.009** |
>   |   espgame ($\gamma$%=250)   |   0.431$\pm$0.007   | 0.467$\pm$0.007 |   0.490$\pm$0.008   | 0.460$\pm$0.006 | **0.501$\pm$0.009** |
>
>
>   [4] Xie, M.-K. and Huang, S.-J. CCMN: A general framework for learning with class-conditional multi-label noise. IEEE Transactions on Pattern Analysis and Machine Intelligence.
>
>   ------
> 3. *What is the baseline model in the ablation study? Do you mean that it learns the prediction model with cross-entropy loss by directly treating all the candidate labels as valid ones?*
>
> * **Response 3**: Thanks to the comments. The baseline model learns the prediction model with cross-entropy loss by directly treating all the candidate labels as valid ones. The only difference between the baseline model and our approach is that our approach performs additional probabilistic graphical disambiguation. Therefore, this baseline model provides a solid basis to evaluate the effectiveness of the probabilistic graphical disambiguation method. We will make it clearer in the revised version.
>
>   ------
> 4. *In formula 6, there is an extra '] '.*
>
> * **Response 4**: Thanks for the reviewer's patience in reading and checking our manuscript. We will carefully proofread the manuscript and correct typos in the revised version.
>
> ------

---

> > ### Comment · Reviewer_uhwx · 2023-08-19
> > **Replying to the authors' rebuttal**
> >
> > Thanks for the detailed rebuttal. The rebuttal has addressed my concerns and I would like to raise my score to positive.

---

> > > ### Author Response · Authors · 2023-08-19
> > > **Thanks**
> > >
> > > Thank you again for the valuable comments and very encouraging feedback. We will check the manuscript again and add the new experimental results in the revised version.

---

### Decision · Program_Chairs · 2023-09-21

**Decision:**

Accept (poster)

**Comment:**

This paper proposes an approach for partial multi-label learning, in which each example can have multiple binary ground truth labels and the training annotations are a superset of the true labels. The paper's approach is to construct a Bayesian network that describes the generative process of the partial labels. The true labels are modeled as latent variables. The paper presents several training techniques to more efficiently optimize a variational lower bound on the data log-likelihood. Experiments show that this approach compares favorably with previously published methods on synthetic and real-world datasets.

The majority of reviewers favored acceptance. They identified as strengths the technical contribution of a novel, principled approach, convincing experiments, and quality of presentation. One reviewer raised a strong concern that the paper had substantial overlap in motivation with "Understanding partial multi-label learning via mutual information," NeurIPS 2021. The other reviewers and the area chair reviewed this reference for similarities. While the other paper applies to the same problem, it takes an approach based on maximizing mutual information between features and identified ground truth labels. This paper instead takes a latent variable approach. While this paper would be strengthened by an empirical comparison with that prior work (in addition to all the other comparisons already done), other reviewers and the area chair did not think that the overlap was that significant.